# BREAKING THE SOFTMAX BOTTLENECK:
# A HIGH-RANK RNN LANGUAGE MODEL

**Zhilin Yang**[*]**, Zihang Dai**[*]**, Ruslan Salakhutdinov, William W. Cohen**
School of Computer Science
Carnegie Mellon University
`{zhiliny,dzihang,rsalakhu,wcohen}@cs.cmu.edu`

## ABSTRACT

We formulate language modeling as a matrix factorization problem, and show that the expressiveness of Softmax-based models (including the majority of neural language models) is limited by a *Softmax bottleneck*. Given that natural language is highly context-dependent, this further implies that in practice Softmax with distributed word embeddings does not have enough capacity to model natural language. We propose a simple and effective method to address this issue, and improve the state-of-the-art perplexities on Penn Treebank and WikiText-2 to 47.69 and 40.68 respectively. The proposed method also excels on the large-scale 1B Word dataset, outperforming the baseline by over 5.6 points in perplexity.[1]

## 1 INTRODUCTION

As a fundamental task in natural language processing, statistical language modeling has gone through significant development from traditional Ngram language models to neural language models in the last decade (Bengio et al., 2003; Mnih & Hinton, 2007; Mikolov et al., 2010). Despite the huge variety of models, as a density estimation problem, language modeling mostly relies on a universal auto-regressive factorization of the joint probability and then models each conditional factor using different approaches. Specifically, given a corpus of tokens $\mathbf{X} = (X_1, \ldots, X_T)$, the joint probability $P(\mathbf{X})$ factorizes as $P(\mathbf{X}) = \prod_t P(X_t \mid X_{<t}) = \prod_t P(X_t \mid C_t)$, where $C_t = X_{<t}$ is referred to as the *context* of the conditional probability hereafter.

Based on the factorization, recurrent neural networks (RNN) based language models achieve state-of-the-art results on various benchmarks (Merity et al., 2017; Melis et al., 2017; Krause et al., 2017). A standard approach is to use a recurrent network to encode the context into a fixed size vector, which is then multiplied by the word embeddings (Inan et al., 2016; Press & Wolf, 2017) using dot product to obtain the logits. The logits are consumed by the Softmax function to give a categorical probability distribution over the next token. In spite of the expressiveness of RNNs as universal approximators (Schäfer & Zimmermann, 2006), an unclear question is whether the combination of dot product and Softmax is capable of modeling the conditional probability, which can vary dramatically with the change of the context.

In this work, we study the expressiveness of the aforementioned Softmax-based recurrent language models from a perspective of matrix factorization. We show that learning a Softmax-based recurrent language model with the standard formulation is essentially equivalent to solving a matrix factorization problem. More importantly, due to the fact that natural language is highly context-dependent, the matrix to be factorized can be high-rank. This further implies that standard Softmax-based language models with distributed (output) word embeddings do not have enough capacity to model natural language. We call this the *Softmax bottleneck*.

We propose a simple and effective method to address the Softmax bottleneck. Specifically, we introduce discrete latent variables into a recurrent language model, and formulate the next-token probability distribution as a *Mixture of Softmaxes* (MoS). Mixture of Softmaxes is more expressive than Softmax and other surrogates considered in prior work. Moreover, we show that MoS learns

---

[*]Equal contribution. Ordering determined by dice rolling.
[1]Code is available at `https://github.com/zihangdai/mos`.

matrices that have much larger normalized singular values and thus much higher rank than Softmax and other baselines on real-world datasets.

We evaluate our proposed approach on standard language modeling benchmarks. MoS substantially improves over the current state-of-the-art results on benchmarks, by up to 3.6 points in terms of perplexity, reaching perplexities 47.69 on Penn Treebank and 40.68 on WikiText-2. We further apply MoS to a dialog dataset and show improved performance over Softmax and other baselines.

Our contribution is two-fold. First, we identify the Softmax bottleneck by formulating language modeling as a matrix factorization problem. Second, we propose a simple and effective method that substantially improves over the current state-of-the-art results.

## 2 LANGUAGE MODELING AS MATRIX FACTORIZATION

As discussed in Section 1, with the autoregressive factorization, language modeling can be reduced to modeling the conditional distribution of the next token $x$ given the context $c$. Though one might argue that a natural language allows an infinite number of contexts due to its compositionality (Pinker, 1994), we proceed with our analysis by considering a finite set of possible contexts. The unboundedness of natural language does not affect our conclusions, which will be discussed later.

We consider a natural language as a finite set of pairs of a context and its conditional next-token distribution[2] $\mathcal{L} = \{(c_1, P^*(X|c_1)), \cdots, (c_N, P^*(X|c_N))\}$, where $N$ is the number of possible contexts. We assume $P^* > 0$ everywhere to account for errors and flexibility in natural language. Let $\{x_1, x_2, \cdots, x_M\}$ denote a set of $M$ possible tokens in the language $\mathcal{L}$. The objective of a language model is to learn a model distribution $P_\theta(X|C)$ parameterized by $\theta$ to match the true data distribution $P^*(X|C)$.

In this work, we study the expressiveness of the parametric model class $P_\theta(X|C)$. In other words, we are asking the following question: given a natural language $\mathcal{L}$, does there exist a parameter $\theta$ such that $P_\theta(X|c) = P^*(X|c)$ for all $c$ in $\mathcal{L}$?

We start by looking at a Softmax-based model class since it is widely used.

### 2.1 SOFTMAX

The majority of parametric language models use a Softmax function operating on a context vector (or hidden state) $\mathbf{h}_c$ and a word embedding $\mathbf{w}_x$ to define the conditional distribution $P_\theta(x|c)$. More specifically, the model distribution is usually written as

$$P_\theta(x|c) = \frac{\exp \mathbf{h}_c^\top \mathbf{w}_x}{\sum_{x'} \exp \mathbf{h}_c^\top \mathbf{w}_{x'}} \tag{1}$$

where $\mathbf{h}_c$ is a function of $c$, and $\mathbf{w}_x$ is a function of $x$. Both functions are parameterized by $\theta$. Both the context vector $\mathbf{h}_c$ and the word embedding $\mathbf{w}_x$ have the same dimension $d$. The dot product $\mathbf{h}_c^\top \mathbf{w}_x$ is called a *logit*.

To help discuss the expressiveness of Softmax, we define three matrices:

$$\mathbf{H}_\theta = \begin{bmatrix} \mathbf{h}_{c_1}^\top \\ \mathbf{h}_{c_2}^\top \\ \cdots \\ \mathbf{h}_{c_N}^\top \end{bmatrix} ; \ \mathbf{W}_\theta = \begin{bmatrix} \mathbf{w}_{x_1}^\top \\ \mathbf{w}_{x_2}^\top \\ \cdots \\ \mathbf{w}_{x_M}^\top \end{bmatrix} ; \ \mathbf{A} = \begin{bmatrix} \log P^*(x_1|c_1), & \log P^*(x_2|c_1) & \cdots & \log P^*(x_M|c_1) \\ \log P^*(x_1|c_2), & \log P^*(x_2|c_2) & \cdots & \log P^*(x_M|c_2) \\ \vdots & \vdots & \ddots & \vdots \\ \log P^*(x_1|c_N), & \log P^*(x_2|c_N) & \cdots & \log P^*(x_M|c_N) \end{bmatrix}$$

where $\mathbf{H}_\theta \in \mathbb{R}^{N \times d}$, $\mathbf{W}_\theta \in \mathbb{R}^{M \times d}$, $\mathbf{A} \in \mathbb{R}^{N \times M}$, and the rows of $\mathbf{H}_\theta$, $\mathbf{W}_\theta$, and $\mathbf{A}$ correspond to context vectors, word embeddings, and log probabilities of the true data distribution respectively. We use the subscript $\theta$ because $(\mathbf{H}_\theta, \mathbf{W}_\theta)$ is effectively a function indexed by the parameter $\theta$, from the joint function family $\mathcal{U}$. Concretely, $\mathbf{H}_\theta$ is implemented as deep neural networks, such as a recurrent network, while $\mathbf{W}_\theta$ is instantiated as an embedding lookup.

We further specify a set of matrices formed by applying *row-wise shift* to $\mathbf{A}$

$$F(\mathbf{A}) = \{\mathbf{A} + \mathbf{\Lambda} \mathbf{J}_{N,M} | \mathbf{\Lambda} \text{ is diagonal and } \mathbf{\Lambda} \in \mathbb{R}^{N \times N}\},$$

---

[2] We use capital letters for variables and small letters for constants.

where $\mathbf{J}_{N,M}$ is an all-ones matrix with size $N \times M$. Essentially, the row-wise shift operation adds an arbitrary real number to each row of $\mathbf{A}$. Thus, $F(\mathbf{A})$ is an infinite set. Notably, the set $F(\mathbf{A})$ has two important properties (see Appendix A for the proof), which are key to our analysis.

**Property 1.** *For any matrix $\mathbf{A}'$, $\mathbf{A}' \in F(\mathbf{A})$ if and only if Softmax$(\mathbf{A}') = P^*$. In other words, $F(\mathbf{A})$ defines the set of **all** possible logits that correspond to the true data distribution.*

**Property 2.** *For any $\mathbf{A}_1 \neq \mathbf{A}_2 \in F(\mathbf{A})$, $|rank(\mathbf{A}_1) - rank(\mathbf{A}_2)| \leq 1$. In other words, all matrices in $F(\mathbf{A})$ have similar ranks, with the maximum rank difference being 1.*

Based on the Property 1 of $F(\mathbf{A})$, we immediately have the following Lemma.

**Lemma 1.** *Given a model parameter $\theta$, $\mathbf{H}_\theta \mathbf{W}_\theta^\top \in F(\mathbf{A})$ if and only if $P_\theta(X|c) = P^*(X|c)$ for all $c$ in $\mathcal{L}$.*

Now the expressiveness question becomes: does there exist a parameter $\theta$ and $\mathbf{A}' \in F(\mathbf{A})$ such that

$$\mathbf{H}_\theta \mathbf{W}_\theta^\top = \mathbf{A}'.$$

This is essentially a matrix factorization problem. We want the model to learn matrices $\mathbf{H}_\theta$ and $\mathbf{W}_\theta$ that are able to factorize some matrix $\mathbf{A}' \in F(\mathbf{A})$. First, note that for a valid factorization to exist, the rank of $\mathbf{H}_\theta \mathbf{W}_\theta^\top$ has to be at least as large as the rank of $\mathbf{A}'$. Further, since $\mathbf{H}_\theta \in \mathbb{R}^{N \times d}$ and $\mathbf{W}_\theta \in \mathbb{R}^{M \times d}$, the rank of $\mathbf{H}_\theta \mathbf{W}_\theta^\top$ is strictly upper bounded by the embedding size $d$. As a result, if $d \geq \text{rank}(\mathbf{A}')$, a universal approximator can theoretically recover $\mathbf{A}'$. However, if $d < \text{rank}(\mathbf{A}')$, no matter how expressive the function family $\mathcal{U}$ is, no $(\mathbf{H}_\theta, \mathbf{W}_\theta)$ can even theoretically recover $\mathbf{A}'$. We summarize the reasoning above as follows (see Appendix A for the proof).

**Proposition 1.** *Given that the function family $\mathcal{U}$ is a universal approximator, there exists a parameter $\theta$ such that $P_\theta(X|c) = P^*(X|c)$ for all $c$ in $\mathcal{L}$ if and only if $d \geq \min_{\mathbf{A}' \in F(\mathbf{A})} rank(\mathbf{A}')$.*

Combining Proposition 1 with the Property 2 of $F(\mathbf{A})$, we are now able to state the *Softmax Bottleneck* problem formally.

**Corollary 1.** *(**Softmax Bottleneck**) If $d < rank(\mathbf{A}) - 1$, for any function family $\mathcal{U}$ and any model parameter $\theta$, there exists a context $c$ in $\mathcal{L}$ such that $P_\theta(X|c) \neq P^*(X|c)$.*

The above corollary indicates that when the dimension $d$ is too small, Softmax does not have the capacity to express the true data distribution. Clearly, this conclusion is not restricted to a finite language $\mathcal{L}$. When $\mathcal{L}$ is infinite, one can always take a finite subset and the Softmax bottleneck still exists. Next, we discuss why the Softmax bottleneck is an issue by presenting our hypothesis that $\mathbf{A}$ is high-rank for natural language.

## 2.2 Hypothesis: Natural Language is High-Rank

We hypothesize that for a natural language $\mathcal{L}$, the log probability matrix $\mathbf{A}$ is a high-rank matrix. It is difficult (if possible) to rigorously prove this hypothesis since we do not have access to the true data distribution of a natural language. However, it is suggested by the following intuitive reasoning and empirical observations:

- Natural language is highly context-dependent (Mikolov & Zweig, 2012). For example, the token "north" is likely to be followed by "korea" or "korean" in a news article on international politics, which however is unlikely in a textbook on U.S. domestic history. We hypothesize that such subtle context dependency should result in a high-rank matrix $\mathbf{A}$.

- If $\mathbf{A}$ is low-rank, it means humans only need a limited number (e.g. a few hundred) of bases, and all semantic meanings can be created by (potentially) negating and (weighted) averaging these bases. However, it is hard to find a natural concept in linguistics and cognitive science that corresponds to such bases, which questions the existence of such bases. For example, semantic meanings might not be those bases since a few hundred meanings may not be enough to cover everyday meanings, not to mention niche meanings in specialized domains.

- Empirically, our high-rank language model outperforms conventional low-rank language models on several benchmarks, as shown in Section 3. We also provide evidences in Section 3.3 to support our hypothesis that learning a high-rank language model is important.

Given the hypothesis that natural language is high-rank, it is clear that the Softmax bottleneck limits the expressiveness of the models. In practice, the embedding dimension $d$ is usually set at the scale of $10^2$, while the rank of $\mathbf{A}$ can possibly be as high as $M$ (at the scale of $10^5$), which is orders of magnitude larger than $d$. Softmax is effectively learning a low-rank approximation to $\mathbf{A}$, and our experiments suggest that such approximation loses the ability to model context dependency, both qualitatively and quantitatively (Cf. Section 3).

## 2.3 EASY FIXES?

Identifying the Softmax bottleneck immediately suggests some possible "easy fixes". First, as considered by a lot of prior work, one can employ a non-parametric model, namely an Ngram model (Kneser & Ney, 1995). Ngram models are not constrained by any parametric forms so it can universally approximate any natural language, given enough parameters. Second, it is possible to increase the dimension $d$ (e.g., to match $M$) so that the model can express a high-rank matrix $\mathbf{A}$.

However, these two methods increase the number of parameters dramatically, compared to using a low-dimensional Softmax. More specifically, an Ngram needs $(N \times M)$ parameters in order to express $\mathbf{A}$, where $N$ is potentially unbounded. Similarly, a high-dimensional Softmax requires $(M \times M)$ parameters for the word embeddings. Increasing the number of model parameters easily leads to overfitting. In past work, Kneser & Ney (1995) used back-off to alleviate overfitting. Moreover, as deep learning models were tuned by extensive hyper-parameter search, increasing the dimension $d$ beyond several hundred is not helpful[3] (Merity et al., 2017; Melis et al., 2017; Krause et al., 2017).

Clearly there is a tradeoff between expressiveness and generalization on language modeling. Naively increasing the expressiveness hurts generalization. Below, we introduce an alternative approach that increases the expressiveness without exploding the parametric space.

## 2.4 MIXTURE OF SOFTMAXES: A HIGH-RANK LANGUAGE MODEL

We propose a high-rank language model called Mixture of Softmaxes (MoS) to alleviate the Softmax bottleneck issue. MoS formulates the conditional distribution as

$$P_\theta(x|c) = \sum_{k=1}^{K} \pi_{c,k} \frac{\exp \mathbf{h}_{c,k}^\top \mathbf{w}_x}{\sum_{x'} \exp \mathbf{h}_{c,k}^\top \mathbf{w}_{x'}}; \quad \text{s.t.} \sum_{k=1}^{K} \pi_{c,k} = 1$$

where $\pi_{c,k}$ is the *prior* or *mixture weight* of the $k$-th component, and $\mathbf{h}_{c,k}$ is the $k$-th context vector associated with context $c$. In other words, MoS computes $K$ Softmax distributions and uses a weighted average of them as the next-token probability distribution. Similar to prior work on recurrent language modeling (Merity et al., 2017; Melis et al., 2017; Krause et al., 2017), we first apply a stack of recurrent layers on top of $\mathbf{X}$ to obtain a sequence of hidden states $(\mathbf{g}_1, \cdots, \mathbf{g}_T)$. The prior and the context vector for context $c_t$ are parameterized as $\pi_{c_t,k} = \frac{\exp \mathbf{w}_{\pi,k}^\top \mathbf{g}_t}{\sum_{k'=1}^{K} \exp \mathbf{w}_{\pi,k'}^\top \mathbf{g}_t}$ and $\mathbf{h}_{c_t,k} = \tanh(\mathbf{W}_{h,k} \mathbf{g}_t)$ where $\mathbf{w}_{\pi,k}$ and $\mathbf{W}_{h,k}$ are model parameters.

Our method is simple and easy to implement, and has the following advantages:

- *Improved expressiveness* (compared to Softmax). MoS is theoretically more (or at least equally) expressive compared to Softmax given the same dimension $d$. This can be seen by the fact that MoS with $K = 1$ is reduced to Softmax. More importantly, MoS effectively approximates $\mathbf{A}$ by

$$\hat{\mathbf{A}}_{\text{MoS}} = \log \sum_{k=1}^{K} \mathbf{\Pi}_k \exp(\mathbf{H}_{\theta,k} \mathbf{W}_\theta^\top)$$

  where $\mathbf{\Pi}_k$ is an $(N \times N)$ diagonal matrix with elements being the prior $\pi_{c,k}$. Because $\hat{\mathbf{A}}_{\text{MoS}}$ is a nonlinear function (*log_sum_exp*) of the context vectors and the word embeddings, $\hat{\mathbf{A}}_{\text{MoS}}$ can be arbitrarily high-rank. As a result, MoS does not suffer from the rank limitation, compared to Softmax.

---

[3]This is also confirmed by our preliminary experiments.

- *Improved generalization* (compared to Ngram). Ngram models and high-dimensional Softmax (Cf. Section 2.3) improve the expressiveness but do not generalize well. In contrast, MoS does not have a generalization issue due to the following reasons. First, MoS defines the following generative process: a discrete latent variable $k$ is first sampled from $\{1, \cdots, K\}$, and then the next token is sampled based on the $k$-th Softmax component. By doing so we introduce an inductive bias that the next token is generated based on a latent discrete decision (e.g., a topic), which is often safe in language modeling (Blei et al., 2003). Second, since $\hat{\mathbf{A}}_{\text{MoS}}$ is defined by a nonlinear function and not restricted by the rank bottleneck, in practice it is possible to reduce $d$ to compensate for the increase of model parameters brought by the mixture structure. As a result, MoS has a similar model size compared to Softmax and thus is not prone to overfitting.

## 2.5 Mixture of Contexts: A Low-Rank Baseline

Another possible approach is to directly mix the context vectors (or logits) before taking the Softmax, rather than mixing the probabilities afterwards as in MoS. Specifically, the conditional distribution is parameterized as

$$P_\theta(x|c) = \frac{\exp\left(\sum_{k=1}^K \pi_{c,k}\mathbf{h}_{c,k}\right)^\top \mathbf{w}_x}{\sum_{x'} \exp\left(\sum_{k=1}^K \pi_{c,k}\mathbf{h}_{c,k}\right)^\top \mathbf{w}_{x'}} = \frac{\exp\left(\sum_{k=1}^K \pi_{c,k}\mathbf{h}_{c,k}^\top \mathbf{w}_x\right)}{\sum_{x'} \exp\left(\sum_{k=1}^K \pi_{c,k}\mathbf{h}_{c,k}^\top \mathbf{w}_{x'}\right)}, \tag{2}$$

where $\mathbf{h}_{c,k}$ and $\pi_{c,k}$ share the same parameterization as in MoS. Despite its superficial similarity to MoS, this model, which we refer to as mixture of contexts (MoC), actually suffers from the same rank limitation problem as Softmax. This can be easily seen by defining $\mathbf{h}'_c = \sum_{k=1}^K \pi_{c,k}\mathbf{h}_{c,k}$, which turns the MoC parameterization (2) into $P_\theta(x|c) = \frac{\exp \mathbf{h}'^\top_c \mathbf{w}_x}{\sum_{x'} \exp \mathbf{h}'^\top_c \mathbf{w}_{x'}}$. Note that this is equivalent to the Softmax parameterization (1). Thus, performing mixture in the feature space can only make the function family $\mathcal{U}$ more expressive, but does not change the fact that the rank of $\mathbf{H}_\theta \mathbf{W}_\theta^\top$ is upper bounded by the embedding dimension $d$. In our experiments, we implement MoC as a baseline and compare it experimentally to MoS.

## 3 Experiments

### 3.1 Main Results

We conduct a series of experiments with the following settings:

- Following previous work (Krause et al., 2017; Merity et al., 2017; Melis et al., 2017), we evaluate the proposed MoS model on two widely used language modeling datasets, namely Penn Treebank (PTB) (Mikolov et al., 2010) and WikiText-2 (WT2) (Merity et al., 2016) based on perplexity. For fair comparison, we closely follow the regularization and optimization techniques introduced by Merity et al. (2017). We heuristically and manually search hyper-parameters for MoS based on the validation performance while limiting the model size (see Appendix B.1 for our hyper-parameters).

- To investigate whether the effectiveness of MoS can be extended to even larger datasets, we conduct an additional language modeling experiment on the 1B Word dataset (Chelba et al., 2013). Specifically, we lower-case the text and choose the top 100K tokens as the vocabulary. A standard neural language model with 2 layers of LSTMs followed by a Softmax output layer is used as the baseline. Again, the network size of MoS is adjusted to ensure a comparable number of parameters. Notably, dropout was not used, since we found it not helpful to either model (see Appendix B.2 for more details).

- To show that the MoS is a generic structure that can be used to model other context-dependent distributions, we additionally conduct experiments in the dialog domain. We use the Switchboard dataset (Godfrey & Holliman, 1997) preprocessed by Zhao et al. (2017)[4] to train a Seq2Seq (Sutskever et al., 2014) model with MoS added to the decoder RNN. Then, a Seq2Seq model using Softmax and another one augmented by MoC with comparable parameter sizes

---

[4]https://github.com/snakeztc/NeuralDialog-CVAE/tree/master/data

| Model | #Param | Validation | Test |
|---|---|---|---|
| Mikolov & Zweig (2012) – RNN-LDA + KN-5 + cache | 9M[‡] | - | 92.0 |
| Zaremba et al. (2014) – LSTM | 20M | 86.2 | 82.7 |
| Gal & Ghahramani (2016) – Variational LSTM (MC) | 20M | - | 78.6 |
| Kim et al. (2016) – CharCNN | 19M | - | 78.9 |
| Merity et al. (2016) – Pointer Sentinel-LSTM | 21M | 72.4 | 70.9 |
| Grave et al. (2016) – LSTM + continuous cache pointer[†] | - | - | 72.1 |
| Inan et al. (2016) – Tied Variational LSTM + augmented loss | 24M | 75.7 | 73.2 |
| Zilly et al. (2016) – Variational RHN | 23M | 67.9 | 65.4 |
| Zoph & Le (2016) – NAS Cell | 25M | - | 64.0 |
| Melis et al. (2017) – 2-layer skip connection LSTM | 24M | 60.9 | 58.3 |
| Merity et al. (2017) – AWD-LSTM w/o finetune | 24M | 60.7 | 58.8 |
| Merity et al. (2017) – AWD-LSTM | 24M | 60.0 | 57.3 |
| Ours – AWD-LSTM-MoS w/o finetune | 22M | 58.08 | 55.97 |
| Ours – AWD-LSTM-MoS | 22M | **56.54** | **54.44** |
| Merity et al. (2017) – AWD-LSTM + continuous cache pointer[†] | 24M | 53.9 | 52.8 |
| Krause et al. (2017) – AWD-LSTM + dynamic evaluation[†] | 24M | 51.6 | 51.1 |
| Ours – AWD-LSTM-MoS + dynamic evaluation[†] | 22M | **48.33** | **47.69** |

Table 1: Single model perplexity on validation and test sets on Penn Treebank. Baseline results are obtained from Merity et al. (2017) and Krause et al. (2017). † indicates using dynamic evaluation.

| Model | #Param | Validation | Test |
|---|---|---|---|
| Inan et al. (2016) – Variational LSTM + augmented loss | 28M | 91.5 | 87.0 |
| Grave et al. (2016) – LSTM + continuous cache pointer[†] | - | - | 68.9 |
| Melis et al. (2017) – 2-layer skip connection LSTM | 24M | 69.1 | 65.9 |
| Merity et al. (2017) – AWD-LSTM w/o finetune | 33M | 69.1 | 66.0 |
| Merity et al. (2017) – AWD-LSTM | 33M | 68.6 | 65.8 |
| Ours – AWD-LSTM-MoS w/o finetune | 35M | 66.01 | 63.33 |
| Ours – AWD-LSTM-MoS | 35M | **63.88** | **61.45** |
| Merity et al. (2017) – AWD-LSTM + continuous cache pointer [†] | 33M | 53.8 | 52.0 |
| Krause et al. (2017) – AWD-LSTM + dynamic evaluation[†] | 33M | 46.4 | 44.3 |
| Ours – AWD-LSTM-MoS + dynamical evaluation[†] | 35M | **42.41** | **40.68** |

Table 2: Single model perplexity over WikiText-2. Baseline results are obtained from Merity et al. (2017) and Krause et al. (2017). † indicates using dynamic evaluation.

are used as baselines. For evaluation, we include both the perplexity and the precision/recall of Smoothed Sentence-level BLEU, as suggested by Zhao et al. (2017). When generating responses, we use beam search with beam size 10, restrict the maximum length to 30, and retain the top-5 responses.

The language modeling results on PTB and WT2 are presented in Table 1 and Table 2 respectively. With a comparable number of parameters, MoS outperforms all baselines with or without dynamic evaluation, and substantially improves over the current state of the art, by up to 3.6 points in perplexity.

| Model | #Param | Train | Validation | Test |
|---|---|---|---|---|
| Softmax | 119M | 41.47 | 43.86 | 42.77 |
| MoS | 113M | **36.39** | **38.01** | **37.10** |

Table 3: Perplexity comparison on 1B word dataset. Train perplexity is the average of the last 4,000 updates.

The improvement on the large-scale dataset is even more significant. As shown in Table 3, MoS outperforms Softmax by over 5.6 points in perplexity. It suggests the effectiveness of MoS is not limited to small datasets where many regularization techniques are used. Note that with limited computational resources, we didn't tune the hyper-parameters for MoS.

| Model | Perplexity | BLEU-1 | | BLEU-2 | | BLEU-3 | | BLEU-4 | |
|---|---|---|---|---|---|---|---|---|---|
| | | prec | recall | prec | recall | prec | recall | prec | recall |
| Seq2Seq-Softmax | 34.657 | 0.249 | 0.188 | 0.193 | 0.151 | 0.168 | 0.133 | 0.141 | 0.111 |
| Seq2Seq-MoC | 33.291 | 0.259 | 0.198 | 0.202 | 0.159 | 0.176 | 0.140 | 0.148 | 0.117 |
| Seq2Seq-MoS | **32.727** | **0.272** | **0.206** | **0.213** | **0.166** | **0.185** | **0.146** | **0.157** | **0.123** |

Table 4: Evaluation scores on Switchboard.

Further, the experimental results on Switchboard are summarized in Table 4[5]. Clearly, on all metrics, MoS outperforms MoC and Softmax, showing its general effectiveness.

## 3.2 ABLATION STUDY

To further verify the improvement shown above does come from the MoS structure rather than adding another hidden layer or finding a particular set of hyper-parameters, we conduct an ablation study on both PTB and WT2. Firstly, we compare MoS with an MoC architecture with the same number of layers, hidden sizes, and embedding sizes, which thus has the same number of parameters. In addition, we adopt the hyper-parameters used to obtain the best MoS model (denoted as MoS hyper-parameters), and train a baseline AWD-LSTM. To avoid distractive factors and save computational resources, all ablative experiments excluded the use of finetuing and dynamic evaluation.

The results are shown in Table 5. Compared to the vanilla AWD-LSTM, though being more expressive, MoC performs only better on PTB, but worse on WT2. It suggests that simply adding another hidden layer or employing a mixture structure in the feature space does not guarantee a better performance. On the other hand, training AWD-LSTM using MoS hyper-parameters severely hurts the performance, which rules out hyper-parameters as the main source of improvement.

| Model | PTB | | WT2 | |
|---|---|---|---|---|
| | **Validation** | **Test** | **Validation** | **Test** |
| AWD-LSTM-MoS | **58.08** | **55.97** | **66.01** | **63.33** |
| AWD-LSTM-MoC | 59.82 | 57.55 | 68.76 | 65.98 |
| AWD-LSTM (Merity et al. (2017) hyper-parameters) | 61.49 | 58.95 | 68.73 | 65.40 |
| AWD-LSTM (MoS hyper-parameters) | 78.86 | 74.86 | 72.73 | 69.18 |

Table 5: Ablation study on Penn Treebank and WikiText-2 without finetuning or dynamical evaluation.

## 3.3 VERIFY THE ROLE OF RANK

While the study above verifies that MoS is the key to achieving the state-of-the-art performance, it is still not clear whether the superiority of MoS comes from its potential high rank, as suggested by our theoretical analysis in Section 2. In the sequel, we take steps to verify this hypothesis.

- Firstly, we verify that MoS does induce a high-rank log-probability matrix empirically, while MoC and Softmax fail. On the validation or test set of PTB with tokens $\mathbf{X} = \{X_1, \ldots, X_T\}$, we compute the log probabilities $\{\log P(X_i \mid X_{<i}) \in \mathbb{R}^M\}_{t=1}^T$ for each token using all three models. Then, for each model, we stack all $T$ log-probability vectors into a $T \times M$ matrix, resulting in $\hat{\mathbf{A}}_{\text{MoS}}$, $\hat{\mathbf{A}}_{\text{MoC}}$ and $\hat{\mathbf{A}}_{\text{Softmax}}$. Theoretically, the number of non-zero singular values of a matrix is equal to its rank. However, performing singular value decomposition of real valued matrices using numerical approaches often encounter roundoff errors. Hence, we adopt the expected roundoff error suggested by Press (2007) when estimating the ranks of $\hat{\mathbf{A}}_{\text{MoS}}$, $\hat{\mathbf{A}}_{\text{MoC}}$ and $\hat{\mathbf{A}}_{\text{Softmax}}$.

  The estimated ranks are shown in Table 6. As predicted by our theoretical analysis, the matrix ranks induced by Softmax and MoC are both limited by the corresponding embedding sizes. By contrast, the matrix rank obtained from MoS does not suffer from this constraint, almost reaching full rank ($M = 10000$). In appendix C.1, we give additional evidences for the higher rank of MoS.

---

[5]The numbers are not directly comparable to Zhao et al. (2017) since their Seq2Seq implementation and evaluation scripts are not publicly available.

| Model | Validation | Test |
|---|---|---|
| Softmax | 400 | 400 |
| MoC | 280 | 280 |
| MoS | **9981** | **9981** |

Table 6: Rank comparison on PTB. To ensure comparable model sizes, the embedding sizes of Softmax, MoC and MoS are 400, 280, 280 respectively. The vocabulary size, i.e., $M$, is 10,000 for all models.

| #Softmax | Rank | Perplexity |
|---|---|---|
| 3 | 6467 | 58.62 |
| 5 | 8930 | 57.36 |
| 10 | 9973 | 56.33 |
| 15 | 9981 | 55.97 |
| 20 | 9981 | 56.17 |

Table 7: Empirical rank and test perplexity on PTB with different number of Softmaxes.

- Secondly, we show that, before reaching full rank, increasing the number of mixture components in MoS also increases the rank of the log-probability matrix, which in turn leads to improved performance (lower perplexity). Specifically, on PTB, with other hyper-parameters fixed as used in section 3.1, we vary the number of mixtures used in MoS and compare the corresponding empirical rank and test perplexity without finetuning. Table 7 summarizes the results. This clear positive correlation between rank and performance strongly supports the our theoretical analysis in section 2. Moreover, note that after reaching almost full rank (i.e., using 15 mixture components), further increasing the number of components degrades the performance due to overfitting (as we inspected the training and test perplexities).

- In addition, as performance improvement can often come from better regularization, we investigate whether MoS has a better, though unexpected, regularization effect compared to Softmax. We consider the 1B word dataset where overfitting is unlikely and no explicit regularization technique (e.g., dropout) is employed. As we can see from the left part of Table 3, MoS and Softmax achieve a similar generalization gap, i.e., the performance gap between the test set and the training set. It suggests both models have similar regularization effects. Meanwhile, MoS has a lower training perplexity compared to Softmax, indicating that the improvement of MoS results from improved expressiveness.

- The last evidence we provide is based on an *inverse* experiment. Empirically, we find that when Softmax does *not* suffer from a rank limitation, e.g., in character-level language modeling, using MoS will *not* improve the performance. Due to lack of space, we refer readers to Appendix C.2 for details.

## 3.4 ADDITIONAL ANALYSIS

**MoS computational time** The expressiveness of MoS does come with a computational cost—computing a $K$-times larger Softmax. To give readers a concrete idea of the influence on training time, we perform detailed analysis in Appendix C.3. As we will see, computational wall time of MoS is actually *sub-linear* w.r.t. the number of Softmaxes $K$. In most settings, we observe a two to three times slowdown when using MoS with up to 15 mixture components.

**Qualitative analysis** Finally, we conduct a case study on PTB to see how MoS improves the next-token prediction in detail. Due to lack of space, we refer readers to Appendix C.4 for details. The key insight from the case study is that MoS is better at making context-dependent predictions. Specifically, given the same immediate preceding word, MoS will produce distinct next-step prediction based on long-term context in history. By contrast, the baseline often yields similar next-step prediction, independent of the long-term context.

## 4 RELATED WORK

In language modeling, Hutchinson et al. (2011; 2012) have previously considered the problem from a matrix rank perspective. However, their focus was to improve the generalization of Ngram language models via a sparse plus low-rank approximation. By contrast, as neural language models already generalize well, we focus on a high-rank neural language model that improves expressiveness without sacrificing generalization. Neubig & Dyer (2016) proposed to mix Ngram and neural language models to unify and benefit from both. However, this mixture might not generalize well since an Ngram model, which has poor generalization, is included. Moreover, the fact that the

two components are separately trained can limit its expressiveness. Levy & Goldberg (2014) also considered the matrix factorization perspective, but in the context of learning word embeddings.

In a general sense, Mixture of Softmaxes proposed in this work can be seen as a particular instantiation of the long-existing idea called Mixture of Experts (MoE) (Jacobs et al., 1991). However, there are two core differences. Firstly, MoE has usually been instantiated as mixture of Gaussians to model data in continuous domains (Jacobs et al., 1991; Graves, 2013; Bazzani et al., 2016). More importantly, the motivation of using the mixture structure is distinct. For Gaussian mixture models, the mixture structure is employed to allow for a parameterized multi-modal distribution. By contrast, Softmax by itself can parameterize a multi-modal distribution, and MoS is introduced to break the Softmax bottleneck as discussed in Section 2.

There has been previous work (Eigen et al., 2013; Shazeer et al., 2017) proposing architectures that can be categorized as instantiations of MoC, since the mixture structure is employed in the feature space.[6] The target of Eigen et al. (2013) is to create a more expressive feed-forward layer through the mixture structure. In comparison, Shazeer et al. (2017) focuses on a sparse gating mechanism also on the feature level, which enables efficient conditional computation and allows the training of a very large neural architecture. In addition to having different motivations from our work, all these MoC variants suffer from the same rank limitation problem as discussed in Section 2.

Finally, several previous works have tried to introduce latent variables into sequence modeling (Bayer & Osendorfer, 2014; Gregor et al., 2015; Chung et al., 2015; Gan et al., 2015; Fraccaro et al., 2016; Chung et al., 2016). Except for (Chung et al., 2016), these structures all define a continuous latent variable for each step of the RNN computation, and rely on the SGVB estimator (Kingma & Welling, 2013) to optimize a variational lower bound of the log-likelihood. Since exact integration is infeasible, these models cannot estimate the likelihood (perplexity) exactly at test time. Moreover, for discrete data, the variational lower bound is usually too loose to yield a competitive approximation compared to standard auto-regressive models. As an exception, Chung et al. (2016) utilizes Bernoulli latent variables to model the hierarchical structure in language, where the Bernoulli sampling is replaced by a thresholding operation at test time to give perplexity estimation.

## 5 CONCLUSIONS

Under the matrix factorization framework, the expressiveness of Softmax-based language models is limited by the dimension of the word embeddings, which is termed as the Softmax bottleneck. Our proposed MoS model improves the expressiveness over Softmax, and at the same time avoids overfitting compared to non-parametric models and naively increasing the word embedding dimensions. Our method improves the current state-of-the-art results on standard benchmarks by a large margin, which in turn justifies our theoretical reasoning: it is important to have a high-rank model for natural language.

### ACKNOWLEDGMENTS

This work was supported by the DARPA award D17AP00001, the Google focused award, and the Nvidia NVAIL award.

---

[6]Although Shazeer et al. (2017) name their architecture as MoE, it is not a standard MoE (Jacobs et al., 1991) and should be classified as MoC under our terminology.

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

## A  PROOFS

**Proof of Property 1**

*Proof.* For any $\mathbf{A}' \in F(\mathbf{A})$, let $P_{\mathbf{A}'}(X|C)$ denote the distribution defined by applying Softmax on the logits given by $\mathbf{A}'$. Consider row $i$ column $j$, by definition any entry in $\mathbf{A}'$ can be expressed as $A'_{ij} = A_{ij} + \Lambda_{ii}$. It follows

$$P_{\mathbf{A}'}(x_j|c_i) = \frac{\exp A'_{ij}}{\sum_k \exp A'_{ik}} = \frac{\exp(A_{ij} + \Lambda_{ii})}{\sum_k \exp(A_{ik} + \Lambda_{ii})} = \frac{\exp A_{ij}}{\sum_k \exp A_{ik}} = P^*(x_j|c_i)$$

For any $\mathbf{A}'' \in \{\mathbf{A}'' \mid \text{Softmax}(\mathbf{A}'') = P^*\}$, for any $i$ and $j$, we have

$$P_{\mathbf{A}''}(x_j|c_i) = P_{\mathbf{A}}(x_j|c_i)$$

It follows that for any $i$, $j$, and $k$,

$$\frac{P_{\mathbf{A}''}(x_j|c_i)}{P_{\mathbf{A}''}(x_k|c_i)} = \frac{\exp A''_{ij}}{\exp A''_{ik}} = \frac{\exp A_{ij}}{\exp A_{ik}} = \frac{P_{\mathbf{A}}(x_j|c_i)}{P_{\mathbf{A}}(x_k|c_i)}$$

As a result,

$$A''_{ij} - A_{ij} = A''_{ik} - A_{ik}$$

This means each row in $\mathbf{A}''$ can be obtained by adding a real number to the corresponding row in $\mathbf{A}$. Therefore, there exists a diagonal matrix $\mathbf{\Lambda} \in \mathbb{R}^{N \times N}$ such that

$$\mathbf{A}'' = \mathbf{A} + \mathbf{\Lambda}\mathbf{J}_{N,M}$$

It follows that $\mathbf{A}'' \in F(\mathbf{A})$. $\qquad\square$

**Proof of Property 2**

*Proof.* For any $\mathbf{A}_1$ and $\mathbf{A}_2$ in $F(\mathbf{A})$, by definition we have $\mathbf{A}_1 = \mathbf{A} + \mathbf{\Lambda}_1\mathbf{J}_{N,M}$, and $\mathbf{A}_2 = \mathbf{A} + \mathbf{\Lambda}_2\mathbf{J}_{N,M}$ where $\mathbf{\Lambda}_1$ and $\mathbf{\Lambda}_2$ are two diagonal matrices. It can be rewritten as

$$\mathbf{A}_1 = \mathbf{A}_2 + (\mathbf{\Lambda}_1 - \mathbf{\Lambda}_2)\mathbf{J}_{N,M}$$

Let $S$ be a maximum set of linearly independent rows in $\mathbf{A}_2$. Let $\mathbf{e}_N$ be an all-ones vector with dimension $N$. The $i$-th row vector $\mathbf{a}_{1,i}$ in $\mathbf{A}_1$ can be written as

$$\mathbf{a}_{1,i} = \mathbf{a}_{2,i} + (\Lambda_{1,ii} - \Lambda_{2,ii})\mathbf{e}_N$$

Because $\mathbf{a}_{2,i}$ is a linear combination of vectors in $S$, $\mathbf{a}_{1,i}$ is a linear combination of vectors in $S \cup \{\mathbf{e}_N\}$. It follows that

$$\text{rank}(\mathbf{A}_1) \leq \text{rank}(\mathbf{A}_2) + 1$$

Similarly, we can derive

$$\text{rank}(\mathbf{A}_2) \leq \text{rank}(\mathbf{A}_1) + 1$$

Therefore,

$$|\text{rank}(\mathbf{A}_1) - \text{rank}(\mathbf{A}_2)| \leq 1$$

$\qquad\square$

**Proof of Proposition 1**

*Proof.* If there exists a parameter $\theta$ such that $P_\theta(X|c) = P^*(X|c)$ for all $c$ in $\mathcal{L}$, by Lemma 1, we have $\mathbf{H}_\theta\mathbf{W}_\theta^\top \in F(\mathbf{A})$. As a result, there exists a matrix $\mathbf{A}' \in F(\mathbf{A})$ such that $\mathbf{H}_\theta\mathbf{W}_\theta^\top = \mathbf{A}'$. Because $\mathbf{H}_\theta$ and $\mathbf{W}_\theta$ are of dimensions $(N \times d)$ and $(M \times d)$ respectively, we have

$$d \geq \text{rank}(\mathbf{A}') \geq \min_{\mathbf{A}'' \in F(\mathbf{A})} \text{rank}(\mathbf{A}'')$$

If $d \geq \min_{\mathbf{A}'' \in F(\mathbf{A})} \text{rank}(\mathbf{A}'')$, there exist matrices $\mathbf{A}' \in F(\mathbf{A})$, $\mathbf{H}' \in \mathbb{R}^{N \times d}$ and $\mathbf{W}' \in \mathbb{R}^{M \times d}$, such that $\mathbf{A}'$ can be factorized as $\mathbf{A}' = \mathbf{H}'\mathbf{W}'^\top$. Because $\mathcal{U}$ is a universal approximator, there exists $\theta$ such that $\mathbf{H}_\theta = \mathbf{H}'$ and $\mathbf{W}_\theta = \mathbf{W}'$. By Lemma 1, $P_\theta(X|c) = P^*(X|c)$ for all $c$ in $\mathcal{L}$. $\qquad\square$

# B Experiment setting and Hyper-parameters

## B.1 PTB and WT2

The hyper-parameters used for MoS in language modeling experiment is summarized below.

| Hyper-parameter | PTB | WT2 |
|---|---|---|
| Learning rate | 20 | 15 |
| Batch size | 12 | 15 |
| Embedding size | 280 | 300 |
| RNN hidden sizes | [960, 960, 620] | [1150,1150,650] |
| Number of mixture components | 15 | 15 |
| Word-level V-dropout | 0.10 | 0.10 |
| Embedding V-dropout | 0.55 | 0.40 |
| Hidden state V-dropout | 0.20 | 0.225 |
| Recurrent weight dropout (Wan et al., 2013) | 0.50 | 0.50 |
| Context vector V-dropout | 0.30 | 0.30 |

Table 8: Hyper-parameters used for MoS. V-dropout abbreviates variational dropout (Gal & Ghahramani, 2016). See (Merity et al., 2017) for more detailed descriptions.

The hyper-parameters used for dynamic evaluation of MoS is summarized below.

| Hyper-parameter | PTB | WT2 |
|---|---|---|
| Batch size | 100 | 100 |
| learning rate ($\eta$) | 0.002 | 0.002 |
| $\epsilon$ | 0.001 | 0.002 |
| $\lambda$ | 0.075 | 0.02 |

Table 9: Hyper-parameters used for dynamic evaluation of MoS. See (Krause et al., 2017) for more detailed descriptions.

## B.2 1B Word Dataset

For training, we use all of the 100 training shards. For validation, we use two shards from the heldout set, namely [heldout-00, heldout-10]. For test, we use another three shards from the heldout set, namely [heldout-20, heldout-30, heldout-40].

The hyper-parameters are listed below.

| Hyper-parameter | Softmax | MoS-7 |
|---|---|---|
| Learning rate | 20 | 20 |
| Batch size | 60 | 60 |
| BPTT langth | 35 | 35 |
| Embedding size | 1024 | 900 |
| RNN hidden sizes | [1024, 1024] | [1024,1024] |
| Dropout rate | 0 | 0 |

Table 10: Hyper-parameters used for Softmax and MoS in experiment on 1B word dataset.

# C Additional experiments

## C.1 Higher empirical rank of MoS compared to MoC and Softmax

In section 3, we compute the rank of different models based on the non-zero singular values of the empirical log-likelihood matrix. Since there can be roundoff mistakes, a less error-prone approach is to directly study the distribution of singular values. Specifically, if more singular values have relatively larger magnitude, the rank of the matrix tends to be higher. Motivated from this intuition,

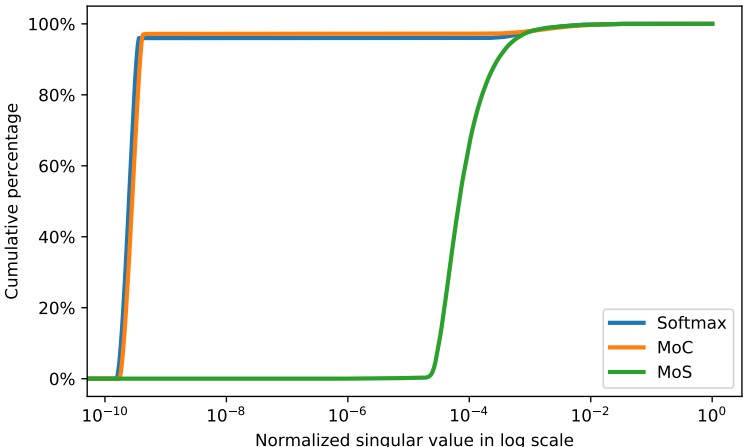

Figure 1: Cumulative percentage of normalized singulars given a value in $[0, 1]$.

we visualize the distribution of the singular values. To account for the different magnitudes of singular values from different models, we first normalize all singular values to $[0, 1]$. Then, we plot the cumulative percentage of normalized singular values, i.e., percentage of normalized singular values below a threshold, in Figure 1. As we can see, most of the singular values of Softmax and MoC concentrate on an area with very low values. In comparison, the concentration area of the MoS singular values is not only several orders larger, but also spans a much wider region. Intuitively, MoS utilizes the corresponding singular vectors to capture a larger and more diverse set of contexts.

| Model | Validation | Test |
|---|---|---|
| Softmax | 4.869 | 4.763 |
| MoC | 4.955 | 4.864 |
| MoS | **5.400** | **5.284** |

Table 11: Empirical expected pairwise KLD on PTB.

What's more, another indicator of high rank is that the model can precisely capture the nuance of difference contexts. If a model can better capture the distinctions among contexts, we expect the next-step conditional distributions to be less similar to each on average. Based on this intuition, we use the expected pairwise Kullback–Leibler divergence (KLD), i.e., $\mathbb{E}_{c,c'\sim\mathcal{C}}\left[\text{KLD}(P(X \mid c)\|P(X \mid c'))\right]$ where $\mathcal{C}$ denotes all possible contexts, as another metric to evaluate the ranks of the three models (MoS, MoC and Softmax). Practically, we sample $c, c'$ from validation or test data of PTB to get the empirical estimations for the three models, which are shown in the right half of Table 11. As we expected, MoS achieves higher expected pairwise KLD, indicating its superiority in covering more contexts of the next-token distribution.

## C.2 AN INVERSE EXPERIMENT ON CHARACTER-LEVEL LANGUAGE MODELING

| Model | | #Param | Train | Validation | Test |
|---|---|---|---|---|---|
| Softmax | (hid1024, emb1024) | 8.42M | 1.35 | 1.41 | 1.49 |
| MoS-7 | (hid910, emb510) | 8.45M | 1.35 | 1.40 | 1.49 |
| MoS-7 | (hid750, emb750) | 8.45M | 1.38 | 1.42 | 1.50 |
| MoS-10 | (hid860, emb452) | 8.43M | 1.35 | 1.41 | 1.49 |
| MoS-10 | (hid683, emb683) | 8.43M | 1.38 | 1.42 | 1.50 |

Table 12: BPC comparison on text8. For MoS, "-$n$" indicates using $n$ mixtures. "hid" and "emb" denote the hidden size and embedding size respectively.

Here, we detail the inverse experiment, which shows that when Softmax does *not* suffer from a rank limitation, using MoS will *not* improve the performance. Notice that character-level language modeling (CharLM) is exactly such a problem, because the rank of the log-likelihood matrix is upper bounded by the vocabulary size, and CharLM usually has a very limited vocabulary (tens of characters). In this case, with the embedding size being hundreds in practice, Softmax is no longer a bottleneck in this task. Hence, we expect MoS to yield similar performance to Softmax on CharLM.

We conduct experiments of CharLM using the text8 dataset (Mahoney, 2011), which consists of 100M characters including only alphabetical characters and spaces derived from Wikipedia. We follow Mikolov et al. (2012) and use the first 90M characters for training, the next 5M for validation and the final 5M for testing. The standard evaluation metric bit-per-character (BPC) is employed. We employ a 1-layer 1024-unit LSTM followed by Softmax as the baseline. For MoS, we consider 7 or 10 mixtures and reduce the hidden and/or embedding size to match the baseline capacity. When decreasing the hidden and/or embedding size, we either keep both the same, or make the hidden size relatively larger. The results are summarized in Table 12. Clearly, the Softmax and MoS obtain the same BPC on the test set and comparable BPC on the validation set, which well match our hypothesis. Since the only difference in word-level language modeling is the existence of the Softmax bottleneck, the distinct behavior of MoS again supports our hypothesis that it is solving the Softmax bottleneck problem.

## C.3 MoS Computational Time

| Model | PTB/bs | PTB/best-1 | WT2/bs | WT2/best-1 | WT2/best-3 | 1B/bs | 1B/best-1 | 1B/best-3 |
|---|---|---|---|---|---|---|---|---|
| Softmax | 1x | 1x | 1x | 1x | 1x | 1x | 1x | 1x |
| MoS-5 | 1.2x | – | 1.3x | – | – | – | – | – |
| MoS-7 | – | – | – | – | – | 3.8x | 5.7x | 2.1x |
| MoS-10 | 1.6x | – | 1.9x | – | – | – | – | – |
| MoS-15 | 1.9x | 2.8x | 2.5x | 6.4x | 2.9x | – | – | – |

Table 13: Training time slowdown compared to Softmax. MoS-$K$ means using $K$ mixture components. "bs" indicates Softmax and MoS use the same batch sizes on one GPU. "best-1" and "best-3" refer to the settings where Softmax and MoS obtain their own best perplexity, with 1 and 3 GPUs respectively.

We evaluate the additional computational cost introduced by MoS. We consider two sets of controlled experiments. In the first set, we compare the training time of MoS and Softmax using the same batch sizes. In the second set, we compare the training time of two methods using the hyperparameter settings that achieve the best performance for each model (i.e., the settings in Tables 1, 2, and 3). In both sets, we control two models to have comparable model sizes.

The results on the three datasets are shown in Table 13. Thanks to the efficiency of matrix multiplication on GPU, the computational wall time of MoS is actually sub-linear w.r.t. the number of Softmaxes $K$. In most settings, we observe a two to three times slowdown when using MoS. Specifically, the "bs" setting measures the computational cost introduced by MoS given enough memory, which is 1.9x, 2.5x, and 3.8x slowdown on PTB, WT2, and 1B respectively. The "best-1" setting is usually slower compared to "bs", because a single batch does not fit into the memory of a single GPU using MoS, in which case we have to split one batch into multiple small ones, resulting in further slowdown. In this sense, the gap between "best-1" and "bs" measures the computational cost introduced due to the increase of memory consumed by MoS. The "best-3" alleviates this issue by using three GPUs, which allows larger-batch training for MoS. In this case, we reduce the computational cost to 2.9x on WT2 and 2.1x on 1B with our best performing model.

Note that the computational cost is closely related to the batch size, which is interleaved with optimization. Though how batch sizes affect optimization remains an open question and might be task dependent, we believe the "best-1" and "best-3" settings well reflect the actual computational cost brought by MoS on language modeling tasks.

## C.4 Qualitative Analysis

Since MoC shows a stronger performance than Softmax on PTB, the qualitative study focuses on the comparison between MoC and MoS. Concretely, given the same context (previous tokens), we search for prediction steps where MoS achieves lower negative log loss than MoC by a margin. We show some representative cases in Table 14 with the following observations:

- Comparing the first two cases, given the same preceding word "N", MoS flexibly adjusts its top predictions based on the different topic quantities being discussed in the context. In comparison, MoC emits quite similar top choices regardless of the context, suggesting its inferiority in make context-dependent predictions.

- In the 3rd case, the context is about international politics, where country/region names are likely to appear. MoS captures this nuance well, and yields top choices that can be used to complete a country name given the immediate preceding word "south". Similarly, in the 4th case, MoS is able to include "ual", a core entity of discussion in the context, in its top predictions. In contrast, MoC gives rather generic predictions irrieselevant to the context in both cases.

- For the 5th and the 6th example, we see MoS is able to exploit less common words accurately according to the context, while MoC fails to yield such choices. This well matches our analysis that MoS has the capacity of modeling context-dependent language.

| #1 Context | managed properly and with a long-term outlook these can become investment-grade quality properties <eos> canadian <unk> production totaled N metric tons in the week ended oct. N up N N from the preceding week 's total of N __?__ | | | |
|---|---|---|---|---|
| MoS top-5 | million 0.38 | **tons 0.24** | billion 0.09 | barrels 0.06 | ounces 0.04 |
| MoC top-5 | billion 0.39 | million 0.36 | trillion 0.05 | <eos> 0.04 | N 0.03 |
| Reference | canadian <unk> production totaled N metric tons in the week ended oct. N up N N from the preceding week 's total of N **tons** statistics canada a federal agency said <eos> | | | |
| #2 Context | the thriving <unk> street area offers <unk> of about $ N a square foot as do <unk> locations along lower fifth avenue <eos> by contrast <unk> in the best retail locations in boston san francisco and chicago rarely top $ N __?__ | | | |
| MoS top-5 | <eos> 0.36 | **a 0.13** | to 0.07 | for 0.07 | and 0.06 |
| MoC top-5 | million 0.39 | billion 0.36 | <eos> 0.05 | to 0.04 | of 0.03 |
| Reference | by contrast <unk> in the best retail locations in boston san francisco and chicago rarely top $ N **a** square foot <eos> | | | |
| #3 Context | as other <unk> governments particularly poland and the soviet union have recently discovered initial steps to open up society can create a momentum for radical change that becomes difficult if not impossible to control <eos> as the days go by the south __?__ | | | |
| MoS top-5 | africa 0.15 | **african 0.15** | <eos> 0.14 | korea 0.08 | korean 0.05 |
| MoC top-5 | <eos> 0.38 | and 0.08 | of 0.06 | or 0.05 | <unk> 0.04 |
| Reference | as the days go by the south **african** government will be ever more hard pressed to justify the continued <unk> of mr. <unk> as well as the continued banning of the anc and enforcement of the state of emergency <eos> | | | |
| #4 Context | shares of ual the parent of united airlines were extremely active all day friday reacting to news and rumors about the proposed $ N billion buy-out of the airline by an <unk> group <eos> wall street 's takeover-stock speculators or risk arbitragers had placed unusually large bets that a takeover would succeed and __?__ | | | |
| MoS top-5 | the 0.14 | that 0.07 | **ual 0.07** | <unk> 0.03 | it 0.02 |
| MoC top-5 | the 0.10 | <unk> 0.06 | that 0.05 | in 0.02 | it 0.02 |
| Reference | wall street 's takeover-stock speculators or risk arbitragers had placed unusually large bets that a takeover would succeed and **ual** stock would rise <eos> | | | |
| #5 Context | the government is watching closely to see if their presence in the <unk> leads to increased <unk> protests and violence if it does pretoria will use this as a reason to keep mr. <unk> behind bars <eos> pretoria has n't forgotten why they were all sentenced to life <unk> in the first place for sabotage and __?__ | | | |
| MoS top-5 | <unk> 0.47 | violence 0.11 | **conspiracy 0.03** | incest 0.03 | civil 0.03 |
| MoC top-5 | <unk> 0.41 | the 0.03 | a 0.02 | other 0.02 | in 0.01 |
| Reference | pretoria has n't forgotten why they were all sentenced to life <unk> in the first place for sabotage and **conspiracy** to <unk> the government <eos> | | | |
| #6 Context | china 's <unk> <unk> program has achieved some successes in <unk> runaway economic growth and stabilizing prices but has failed to eliminate serious defects in state planning and an <unk> drain on state budgets <eos> the official china daily said retail prices of <unk> foods have n't risen since last december but acknowledged that huge government __?__ | | | |
| MoS top-5 | **subsidies 0.15** | spending 0.08 | officials 0.04 | costs 0.04 | <unk> 0.03 |
| MoC top-5 | officials 0.04 | figures 0.03 | efforts 0.03 | <unk> 0.03 | costs 0.03 |
| Reference | the official china daily said retail prices of <unk> foods have n't risen since last december but acknowledged that huge government **subsidies** were a main factor in keeping prices down <eos> | | | |

Table 14: Compaison of next-token prediction on Penn Treebank test data. N stands for a number as the result of preprocessing (Mikolov et al., 2010). The context shown only includes the previous sentence and the current sentence the prediction step resides in.

