# OpenReview forum: "Breaking the Softmax Bottleneck: A High-Rank RNN Language Model"
_ICLR.cc/2018/Conference — Accept (Oral)_

### Official Review · AnonReviewer2 · 2017-11-25
**The authors identify an issue with current prediction models and propose a solution in the form of a mixture of softmax. The idea is interesting, but would require more justification and experimental validation.**

**Rating:** 7
**Confidence:** 4

**Review:**

The authors argue in this paper that due to the limited rank of the  context-to-vocabulary logit matrix in the currently used version of the softmax output layer, it is not able to capture the full complexity of language. As a result, they propose to use a mixture of softmax output layers instead where the mixing probabilities are context-dependent, which allows to obtain a full rank logit matrix in complexity linear in the number of mixture components (here 15). This leads to improvements in the word-level perplexities of the PTB and wikitext2 data sets, and Switchboard BLEU scores.

The question of the expressiveness of the softmax layer, as well as its suitability for word-level prediction, is indeed an important one which has received too little attention. This makes a lot of the questions asked in this paper extremely relevant to the field. However, it is unclear that the rank of the logit matrix is the right quantity to consider. For example, it is easy to describe a rank D NxM matrix where up to 2^D lines have max values at different indices. Further, the first two "observations" in Section 2.2 would be more accurately described as "intuitions" of the authors. As they write themselves "there is no evidence showing that semantic meanings are fully linearly correlated." Why then try to link "meanings" to basis vectors for the rows of A?

To be clear, the proposed model is undoubtedly more expressive than a regular softmax, and although it does come at a substantial computational cost (a back-of-the envelope calculation tells us that computing 15 components of 280d MoS takes the same number of operations as one with dimension 1084 = sqrt (280*280*15)), it apparently manages not to drastically increase overfitting, which is significant.

Unfortunately, this is only tested on relatively small data sets, up to 2M tokens and a vocabulary of size 30K for language modeling. They do constitute a good starting place to test a model, but given the importance of regularization on those specific tasks, it is difficult to predict how the MoS would behave if more training data were available, and if one could e.g. simply try a 1084 dimension embedding for the softmax without having to worry about overfitting.

Another important missing experiment would consist in varying the number of mixture components (this could very well be done on WikiText2). This could help validate the hypothesis: how does the estimated rank vary with the number of components? How about the performance and pairwise KL divergence?

This paper offers a promising direction for language modeling research, but would require more justification, or at least a more developed experimental section.

Pros:
- Important starting question
- Thought-provoking approach
- Experimental gains on small data sets

Cons:
- The link between the intuition and reality of the gains is not obvious
- Experiments limited to small data sets, some obvious questions remain

---

> ### Author Response · Authors · 2017-12-25
> **response**
>
> Thank you for the valuable feedback.
>
> [[Rank and meanings]] It could be possible that A is low-rank for a natural language as it is hard to rule out this possibility rigorously, but we hypothesize that A is high-rank. Our hypothesis is supported by our intuitive reasoning and empirical experiments. Empirically, we give three more pieces of evidences supporting our hypothesis that the rank is the key bottleneck of Softmax and MoS improves the performance by solving the rank bottleneck (section 3.2):
>     - Before the rank saturates to the full rank, using more mixture components in MoS continues to increase the rank of the log-probability matrix. Further, when the rank increases, the perplexity also decreases.
>     - MoS has a similar generalization gap compared to Softmax, which rules out the concern that the improvement actually comes from some unexpected regularization effects of MoS.
>     - In character-level language modeling, since the largest possible rank is upper bounded by the limited vocabulary size, Softmax does not suffer from the rank bottleneck. In this case, Softmax and MoS have almost the same performance, which matches our analysis.
> We agree that linking semantic meanings to bases lacks rigor and this would be better described as intuitions. We have made corresponding changes in the paper.
>
> [[Computation vs. Capacity]]: It is true that MoS involves a larger amount of computation compared to the standard Softmax. However, [Collins et al] suggests the capacity of neural language models is mostly related to the number of parameters, rather than computation. Moreover, powerful models often require a larger amount of computation. For example, the attention based seq2seq model involves much more computation compared to the vanilla seq2seq.
>
> [[Large-scale experiment]]: We have added a “large-scale language modeling experiment” using the 1B Word Dataset (section 3.1), where MoS significantly outperforms the baseline model with a large margin. This indicates that MoS consistently outperforms Softmax, regardless of the scale of the dataset. Also, note that PTB and WT2 are two de-facto benchmarks widely used in previous work on language modeling. None of the following papers had experiments on datasets larger than WT2: Zoph & Le ICLR 2017, Zilly et al ICML 2017, Inan et al ICLR 2017, Grave et al ICLR 2017, Merity et al ICLR 2017.
>
> [[Varying the number of mixtures]]: Thanks for the suggestion. We performed this experiment, whose result is summarized in the second bullet point of section 3.2 (updated version). As expected, the number of mixture components is positively correlated with the empirical rank. More importantly, before the rank saturates to the full rank, MoS with a higher rank leads to a better performance (lower perplexity).
>
> ------------------------------------------------------------------------------------------------------------------------
> [Collins et al] Capacity and Trainability in Recurrent Neural Networks

---

> > ### Comment · AnonReviewer2 · 2018-01-12
> > **post-response**
> >
> > The authors have added some important experiments whose results support their claim, and I do believe that the current version of the paper makes a stronger case. I am still not satisfied that the rank explanation fully captures what is going on there, although it certainly correlates with better results, but this paper will provide an important data point for future research and I am raising my score from 5 to 7.

---

### Official Review · AnonReviewer3 · 2017-11-27
**The paper is grounded on a solid theoretical motivation and the analysis is sound and quite interesting.**

**Rating:** 7
**Confidence:** 5

**Review:**

The authors has addressed my concerns, so I raised my rating.

The paper is grounded on a solid theoretical motivation and the analysis is sound and quite interesting.

There are no results on large corpora such as 1 billion tokens benchmark corpus, or at least medium level corpus with 50 million tokens. The corpora the authors choose are quite small, the variance of the estimates are high, and similar conclusions might not be valid on a large corpus.

[1] provides the results of character level language models on Enwik8 dataset, which shows regularization doesn't have much effect and needs less tuning. Results on this data might be more convincing.

The results of MOS is very good, but the computation complexity is much higher than other baselines. In the experiments, the embedding dimension of MOS is slightly smaller, but the number of mixture is 15. This will make it less usable, I think it's necessary to provide the training time comparison.

Finally experiments on machine translation or speech recognition should be done and to see what improvements the proposed method could bring for BLEU or WER.

[1] Melis, Gábor, Chris Dyer, and Phil Blunsom. "On the state of the art of evaluation in neural language models." arXiv preprint arXiv:1707.05589 (2017).

[2] Joris Pelemans, Noam Shazeer, Ciprian Chelba, Sparse Non-negative Matrix Language Modeling,  Transactions of the Association for Computational Linguistics, vol. 4 (2016), pp. 329-342

[3] Shazeer et al. (2017). Outrageously Large Neural Networks: The Sparsely-Gated Mixture-of-Experts Layer. ICLR 2017

---

> ### Public Comment · (anonymous) · 2017-12-06
> **Re: enwik8**
>
> It's true that regularization doesn't play as important a role with large datasets as it does with small datasets, but enwik8 is character based and it's unclear whether this paper's arguments would apply. Sticking to word level corpora, I'd much sooner recommend Wikitext-103 than the Billion Word corpus which has issues.
>
> Furthermore, language modelling improvements are interesting in their own right without having to validate them via MT or speech recognition.

---

> ### Author Response · Authors · 2017-12-25
> **response**
>
> Thanks for your valuable comments.
>
> [[Large-scale experiment]]: We’ve added a “large-scale language modeling experiment” using the 1B Word Dataset (section 3.1), where MoS significantly outperforms the baseline model by a large margin. This indicates that MoS consistently outperforms Softmax, regardless of the scale of the dataset. Also, note that PTB and WT2 are two de-facto benchmarks widely used in previous work on language modeling. None of the following papers had experiments on datasets larger than WT2: Zoph & Le ICLR 2017, Zilly et al ICML 2017, Inan et al ICLR 2017, Grave et al ICLR 2017, Merity et al ICLR 2017.
>
> [[Character-level LM]]: Firstly, note that the largest possible rank of the log-probability matrix is upper bounded by the vocabulary size. In character-level LM, the vocabulary size is usually much smaller than the embedding size. In this case, Softmax does not suffer from the rank bottleneck problem, and we expect MoS and Softmax to achieve similar performance in practice. To verify our expectation, we perform character-level LM experiment on the text8 dataset, where MoS and Softmax indeed achieve almost the same performance (section 3.2 & appendix C.2 in the updated version).
>
> [[Training time]]: We have added the training time analysis for MoS and provided empirical numbers in the updated versions of the paper (section 3.3 & Appendix C.3). In general, computational wall time of MoS is actually sub-linear w.r.t. the number of mixture components. In most settings, we observe a two to three times slowdown compared to Softmax when using up to 15 components for MoS. We believe such additional computational cost is acceptable for the following reasons:
>     - MoS is highly parallelizable, meaning that using more machines can always speed up the computation almost linearly.
>     - The field of deep learning systems (both hardware and software) is making rapid progress. It might be possible to further optimize MoS on GPUs for fast computation. More developed hardware systems would also further reduce the computational cost.
>     - Historically, important techniques sometimes come with an additional computational cost, e.g., LSTM, attention, deep ResNets. We believe that with MoS, the extra cost is reasonable and the gain is substantial.
>
> [[Application to MT/ASR]]: We believe this is best left to future research, as performing rigorous experiments and careful comparison for such a real-world applications is non-trivial. And we believe language modeling is of its own importance already.

---

### Official Review · AnonReviewer1 · 2017-11-30
**nice idea**

**Rating:** 8
**Confidence:** 4

**Review:**

Language models are important components to many NLP tasks. The current state-of-the-art language models are based on recurrent neural networks which compute the probability of a word given all previous words using a softmax function over a linear function of the RNN's hidden state. This paper argues the softmax is not expressive enough and proposes to use a more flexible mixture of softmaxes. The use of a mixture of softmaxes is motivated from a theoretical point of view by translating language modeling into matrix factorization.

Pros:
--The paper is very well written and easy to follow. The ideas build up on each other in an intuitive way.
--The idea behind the paper is novel: translating language modeling into a matrix factorization problem is new as far as I know.
--The maths is very rigorous.
--The experiment section is thorough.

Cons:
--To claim SOTA all models need to be given the same capacity (same number of parameters). In Table 2 the baselines have a lower capacity. This is an unfair comparison
--I suspect the proposed approach is slower than the baselines. There is no mention of computational cost. Reporting that would help interpret the numbers.

The SOTA claim might not hold if baselines are given the same capacity. But regardless of this, the paper has very strong contributions and deserves acceptance at ICLR.

---

> ### Author Response · Authors · 2017-12-25
> **response**
>
> Thank you for the valuable comments.
>
> [[Claim of SOTA]]: We believe the 2M difference in the number of parameters is negligible compared to the number of parameters we use (i.e., 35M). In fact, we ran MoS in another setting with 31M parameters and got 63.59 on WT2 without finetuning, compared to 63.33 obtained by our best-performing model.
>
> [[Training time]]: Thanks for the suggestion and we have added the training time analysis for MoS and provided empirical numbers in the updated versions of the paper (section 3.3 & Appendix C.3). In general, computational wall time of MoS is actually sub-linear w.r.t. the number of mixture components. In most settings, we observe a two to three times slowdown compared to Softmax when using up to 15 components for MoS. We believe such additional computational cost is acceptable for the following reasons:
>     - MoS is highly parallelizable, meaning that using more machines can always speed up the computation almost linearly.
>     - The field of deep learning systems (both hardware and software) is making rapid progress. It might be possible to further optimize MoS on GPUs for fast computation. More developed hardware systems would also further reduce the computational cost.
>     - Historically, important techniques sometimes come with an additional computational cost, e.g., LSTM, attention, deep ResNets. We believe that with MoS, the extra cost is reasonable and the gain is substantial.

---

### Public Comment · ~Aaron_Jaech1 · 2017-10-30
**prior work on the low-rank problem**

This seems like an interesting idea and a great result!

When I was reading what you wrote about potential easy fixes, I was reminded of the work from Brian Hutchinson where he shows how back-off smoothing is a low-rank approximation of the "A" matrix. (See, "Low Rank Language Models for Small Training Sets" by Hutchinson, Ostendorf, and Fazel.)  In later work by the same authors they use a sparse plus low-rank model to remedy the deficiencies of the low-rank approximation.  (See, "A Sparse Plus Low Rank Maximum Entropy Language Model.") That line of work ended up being impractical for large datasets and the solution that you have come up with seems much more promising.

Another paper that I think is related to your Mixture of Softmaxes is the Mixture of Distributions model from Neubig and Dyer's paper "Generalizing and Hybridizing Count-based and Neural Language Models." They are talking about mixing n-gram distributions with a softmax distribution. Their model is another way of "breaking the softmax bottleneck" although they don't motivate it as such.

---

> ### Author Response · Authors · 2017-11-01
> **response**
>
> Thank you for the comments. We will include the related papers in our later version. For now, we will summarize the difference between the mentioned work and ours as follows:
>
> 1. As discussed in Section 2.3, there is a tradeoff between expressiveness and generalization. Ngram models are expressive but do not generalize well due to data sparsity (or curse of dimensionality). Hutchinson et al. attempted to improve generalization using sparse plus low-rank Ngram models. By contrast, neural language models with standard Softmax generalize well but do not have enough expressiveness (as shown in Sections 2.1 and 2.2). This motivates our high-rank approach that improves expressiveness without sacrificing generalization. In a nutshell, the difference is that we aim to improve expressiveness while Hutchinson et al. aimed to improve generalization, and such a difference is a result of the intrinsic difference between neural and Ngram language models.
>
> 2. There are two key differences between the MoD model (Neubig and Dyer) and ours.
>         - Firstly, the motivations are totally different. Neubig and Dyer proposed to hybridize Ngram and neural language models to unify and benefit from both. In comparison, we identify the Softmax bottleneck problem using a matrix factorization formulation, and hence motivate our approach to break the bottleneck.
>         - Secondly, our approach is end-to-end and achieves a good tradeoff between generalization and expressiveness (Cf. Section 2.3). In comparison, MoD might not generalize well since Ngram models, which have poor generalization, are included in the mixture. Moreover, the Ngram parameters are fixed in MoD, which limits its expressiveness.
> Despite the differences, note that it is possible to combine our work with theirs to further improve language modeling.

---

> > ### Public Comment · ~Aaron_Jaech1 · 2017-11-01
> > **thanks for your response**
> >
> > Thanks for your response. Just so it's clear, I want to reiterate that I really liked your paper and thought it was an original and worthwhile contribution.

---

### Public Comment · (anonymous) · 2017-11-13
**Worse performance of the baseline (AWD-LSTM) under MoS hyperparameters**

In ablation studies, during comparison between MoS and the baseline (AWD-LSTM), the authors use the hyperparameters tuned on MoS for the evaluation of AWD-LSTM. It is claimed that the worse performance of AWD-LSTM under these hyperparameters implies that the better performance of MoS is not due to hyperparamters. I think a fairer comparison would be to search for the best hyperparameters individually for each task. Can the authors elaborate on the motivation for this approach? Also, was an evaluation of MoS performed using the AWD-LSTM hyperparameters provided by Merity et al 2017, and only tuning the MoS specific hyperparameters on top of that? I believe this would be a fairer comparison (if finding the best hyperparameters for each model individually using a comparable extensive grid search is too expensive).

---

> ### Author Response · Authors · 2017-11-13
> **response**
>
> Thanks for your comments. We believe our comparison is fair and MoS is indeed better than the baseline, with the following reasons:
>
> Firstly, the hyper-parameters for MoS are chosen by trial and error (i.e., graduate student descent) rather than an extensive hyper-parameter search such as the one used by [Melis et al]. The baseline AWD-LSTM uses a similar strategy in searching for hyper-parameters. Therefore, both MoS and the baseline are well tuned and comparable.
>
> Secondly, the baseline was the previous state-of-the-art (SOTA) (before our paper is released). It is usually true that one would not expect hyper-parameter tuning alone to substantially improve SOTA results on widely-studied benchmarks.
>
> Thirdly, since MoS introduces another hidden layer, the number of parameters would significantly increase if we kept the embedding size and hidden size the same as the baseline, which would lead to an unfair comparison [Collins et al]. Thus, we have to trim the network size and modify related hyper-parameters accordingly.
>
> Fourthly, the fact that the baseline with MoS hyper-parameters is worse than MoS is just a necessary condition of our argument that MoS is better than the baseline. (And we do not claim it is sufficient; sufficiency is proved by comparison with SOTA).
>
> [Melis et al] On the State of the Art of Evaluation in Neural Language Models
> [Collins et al] Capacity and Trainability in Recurrent Neural Networks

---

> > ### Public Comment · (anonymous) · 2017-11-24
> > **thanks for your response**
> >
> > Thank you for your extensive response.
> >
> > I fully agree with the fact that the baseline model's performance using MoS hyper-parameters is worse than MoS is a necessary condition. I wrongly assumed, that you also find that to be sufficient condition. I concluded that based on this sentence from your paper 'On the other hand, training AWD-LSTM using MoS hyper-parameters severely hurts the performance, which rules out hyper-parameters as the main source of improvement'. It is probably just an unfortunate sentence and you may want to paraphrase it.
> >
> > I found your arguments very persuasive. My only concern is that you use very small batch sizes that make training procedure very slow (is it more than a week on a single GPU for WT2?). Have you tried to train with batch sizes comparable with state-of-the-art model that you compare with?
> >
> > Thank you again for making your statements more clear to me.

---

> > > ### Author Response · Authors · 2017-11-24
> > > **response**
> > >
> > > We haven’t tried larger batch sizes since it does not fit into the memory. We did try training the baseline models with batch size 20 (compared to 40 in the original paper) on Penn Treebank, and the performance degrades a little bit (from 58.95 to 59.10), which indicates that using small batch sizes does not improve the baseline model. The AWD-LSTM paper also confirmed that “relatively large batch sizes (e.g., 40-80) performed better than smaller sizes (e.g., 10-20) for NT-ASGD”. Thus it is likely that MoS will perform even better with larger batch sizes.
> > >
> > > On Penn Treebank, we also tried using lr=20 while keeping batch_size=40 for Softmax, since lr=20 is used for MoS. It turned out this significantly worsened the perplexity of Softmax from 58.95 to 61.39. Combined with the previous analysis, it suggests the performance improvement of MoS does not come from a better choice of learning rate and/or batch size.
> > >
> > > Moreover, in our preliminary experiments, we compared MoS with Softmax using the PyTorch demo code (https://github.com/pytorch/examples/tree/master/word_language_model). MoS (nhid=1059, dropout=0.55, n_softmax=20) obtains a perplexity of 68.04, compared to 72.30 obtained by Softmax (nhid=1500, dropout=0.65). In this experiment, the batch sizes and learning rates are the same and we are again seeing clear gains. Note that we reduced nhid to obtain comparable model sizes, and reduced dropout since the hidden unit size is smaller in our case.

---

### Public Comment · ~Gábor_Melis1 · 2017-12-05
**Training fit?**

How much does training perplexity improve with MoS compared to the baseline model?

If it does improve substantially, that gives more credence to the rank argument.
If it doesn't, then we might be observing unexpected regularization effects.

---

> ### Author Response · Authors · 2017-12-25
> **response**
>
> Thank you for your valuable feedback. We believe that, on large datasets where explicit regularization techniques like dropout are not crucial, the improvement on training perplexity does give information about whether MoS has an unintended regularization effect that can improve the performance.
>
> Thus, in our updated version, we conduct an experiment on the 1B Word dataset, where no dropout or other regularization technique is used. As described in the third bullet point of section 3.2, in this regularization free setting, MoS and Softmax have the same generalization gap (i.e., the gap between training and test error), and performance improvement is fully reflected on the training perplexity. Hence, the superiority of MoS is not caused by some unexpected regularization but improved expressiveness.
>
> We also provide additional evidence in section 3.2 (updated version) to support our theory that achieving a higher rank is the key to the excellence of MoS.

---

> > ### Public Comment · ~Gábor_Melis1 · 2018-01-04
> > **Re: response**
> >
> > Thank you for the update that improved an already very good paper.
> >
> > Informal Rating: 8
> > Confidence: 4

---

### Author Response · Authors · 2017-12-25
**Update to the paper**

To provide faithful answers to the reviews and comments, we have conducted a series of additional experiments and updated the paper accordingly. The core changes are summarized as follows:
    (1) we add a “large-scale language modeling experiment” using the 1B Word Dataset (section 3.1)
    (2) we give three more pieces of “evidence supporting our theory” that the rank is the key bottleneck of Softmax and MoS improves the performance by breaking the rank bottleneck (section 3.2):
        - Empirically, before the rank saturates to the full rank, using more mixture components in MoS continues to increase the rank of the log-probability matrix. Further, the higher the rank is, the lower the perplexity that can be achieved.
        - MoS has a similar generalization gap compared to Softmax, which rules out the concern that the improvement actually comes from some unexpected regularization effects of MoS.
        - In character-level language modeling, since the largest possible rank is upper bounded by the limited vocabulary size, Softmax does not suffer from the rank bottleneck. In this case, Softmax and MoS have almost the same performance, which matches our analysis.
    (3) we perform “training time analysis” for MoS and provide empirical training time comparison (section 3.3 & Appendix C.3)

---

### Public Comment · ~Stephen_Merity1 · 2018-01-03
**Important paper that analyses and addresses a fundamental issue with large vocabularies**

I would strongly recommend this paper be accepted for publication. This paper uncovers a fundamental issue with large vocabularies and goes beyond just analyzing the issue by proposing a helpful method of addressing this. Whilst I was already excited by the initial version of this paper, the follow up work that has been done by the authors is even more informative. Understanding and considering the rank bottlenecks of our models seems an important consideration for future models.
If I can answer any follow up questions in support of this paper I would be happy to.

Informal Rating: 8
Confidence: 5
(work directly in this field and have recreated many aspects of the results since publication)

---

### Public Comment · ~Mitchel_Weintraub1 · 2018-01-26
**Similar algorithm proposed at JHU workshop in 1995**

See the paper from JHU 1995 at:
https://www.researchgate.net/profile/Mitchel_Weintraub/publication/246170642_Fast_Training_and_Portability/links/57505b7008aefe968db72bef/Fast-Training-and-Portability.pdf
See section 3, pp. 6-8.  This describes the factoring of a LM into a mixture of tied multinomials.  The mixture weight computation is slightly different, but the factoring of the overall LM distribution into a set of tied distributions was presented at this workshop.

---

> ### Author Response · Authors · 2018-01-26
> **response**
>
> Thanks for pointing out this related piece we’ve missed. Salute!
>
> We would like to clarify that using a mixture structure is by no means a new idea, as we have noted in Related Work. Instead, the insight on model expressiveness, the integration with modern architectures and optimization algorithms, the SOTA performance, and the consistency between theory and practice, are our foci.
>
> Moreover, there’s an essential difference technically. In the reference, P(w | h) = sum_m P(w|m) P(m|h), while in MoS, P(w | h) = sum_m P(w|m,h) P(m|h). In other words, each mixture component is independent of the history in the reference model, while MoS explicitly models such dependency, making each component much more powerful.

---

### Decision · Program_Chairs · 2018-01-29
**ICLR 2018 Conference Acceptance Decision**

**Decision:**

Accept (Oral)

**Comment:**

Viewing language modeling as a matrix factorization problem, the authors argue that the low rank of word embeddings used by such models limits their expressivity and show that replacing the softmax in such models with a mixture of softmaxes provides an effective way of overcoming this bottleneck. This is an interesting and well-executed paper that provides potentially important insight. It would be good to at least mention prior work related to the language modeling as matrix factorization perspective (e.g. Levy & Goldberg, 2014).